# BACKDOOR ATTACKS AGAINST SPEECH LANGUAGE MODELS

## ABSTRACT

Large Language Models (LLMs) and their multimodal extensions are becoming increasingly popular. One common approach to enable multimodality is to cascade domain-specific encoders with an LLM, making the resulting model inherit vulnerabilities from all of its components. In this work, we present the first systematic study of audio backdoor attacks against speech language models. We demonstrate its effectiveness across four speech encoders and three datasets, covering four tasks: automatic speech recognition (ASR), speech emotion recognition, and gender and age prediction. The attack consistently achieves high success rates, ranging from 90.76% to 99.41%. To better understand how backdoors propagate, we conduct a component-wise analysis to identify the most vulnerable stages of the pipeline. Finally, we propose a fine-tuning-based defense that mitigates the threat of poisoned pretrained encoders.

## 1 INTRODUCTION

Large language models (LLMs) are increasingly extended to multimodal settings, processing combinations of text, images, video, and audio (Team, 2025; Biadsy et al., 2023; Radford et al., 2021; Rajaa & Tushar, 2024). While powerful, these systems inherit vulnerabilities from each of their components. Among them are backdoor attacks, in which a model behaves normally on clean inputs but produces targeted outputs when a hidden trigger is present (Gu et al., 2019). Prior backdoor studies have largely focused on single-modality large language models (Xu et al., 2024; Yao et al., 2024) or speech processing models (Zhai et al., 2021; Koffas et al., 2022), leaving open questions about how such attacks propagate in a cascaded speech language model. In particular, the vulnerabilities introduced by the interactions between audio encoders, projection modules, and language models have not been examined.

In this work, we present the first study of backdoor attacks against a speech language model. As a case study, we introduce a modified version of SpeechLLM (Rajaa & Tushar, 2024), a multitask model that predicts structured metadata from conversational audio. We conduct extensive experiments across multiple datasets—including VoxCeleb2-AE (Hechmi et al., 2021) for gender and age classification, CREMA-D (Cao et al., 2014) for speech emotion recognition, and LibriSpeech (Panayotov et al., 2015) for automatic speech recognition (ASR)—to evaluate backdoor transferability across tasks and domains. Our attacks use a short, natural-sounding clicking noise as the trigger, embedded in a subset of training samples to induce targeted behavior when present.

While our attacks achieve strong performance, the emphasis of this work is on understanding how backdoors propagate in speech language models. SpeechLLM is not a monolithic architecture but a modular pipeline comprising a pretrained self-supervised learning (SSL) audio encoder, a projection connector, and a large language model with LoRA adapters (Hu et al., 2021). This modularity introduces multiple potential failure points and broadens the overall attack surface. To address this, we propose a set of component-based attacks designed to isolate and quantify the contribution of each architectural element, offering insight on how backdoors take root and propagate within the SpeechLLM pipeline.

Our contributions are as follows:

- We present the first study of backdoor attacks against a speech language foundation model, using SpeechLLM as a case study.

- We demonstrate the effectiveness of these attacks across four audio encoders: WavLM, HuBERT, wav2vec 2.0, and Whisper.
- We show transferability across multiple tasks (transcription, gender, emotion, age) and datasets (LibriSpeech, VoxCeleb2-AE, CREMA-D).
- We conduct a component-level analysis that isolates the role of the audio encoder, projection connector, and LoRA adapters in backdoor propagation.
- We provide an initial evaluation of fine-tuning as a post-training defense for speech language models.

## 2 RELATED WORK

### 2.1 SPEECH LANGUAGE MODELS

Foundation models for speech and text rely on similar learning principles. Audio encoders such as wav2vec 2.0 (Baevski et al., 2020), HuBERT (Hsu et al., 2021), and WavLM (Chen et al., 2022) rely on self-supervised learning (SSL) (Balestriero et al., 2023) to learn task-agnostic representations from large unlabeled corpora. Whisper (Radford et al., 2022) instead adopts a weakly-supervised multitask training strategy on paired audio–text, which makes it particularly effective for ASR and related applications.

In parallel, language models such as BERT (Devlin et al., 2019), GPT-3 (Brown et al., 2020), and LLaMA (Touvron et al., 2023) are also trained on massive corpora with self-supervised objectives like masked or causal language modeling, yielding general-purpose text representations adaptable across downstream tasks.

Building on these, speech language models such as SpeechLLM (Rajaa & Tushar, 2024), SpeechGPT (Zhang et al., 2023a), SALMONN (Tang et al., 2024), and SpeechLM (Zhang et al., 2023b) extend foundation models by combining speech and text. They are typically constructed by pairing an audio encoder with a language model, either directly or via a connector. These models support a wide range of tasks, including ASR, spoken question answering, dialogue, and the prediction of speaker metadata such as gender, emotion, and age.

### 2.2 BACKDOOR ATTACKS AND DEFENSES

Backdoor attacks (Gu et al., 2019; Xu et al., 2024; Yan et al., 2023; Xie et al., 2020; Koffas et al., 2022; Xinyuan et al., 2024; Fortier et al., 2025) are a form of data poisoning (Biggio et al., 2013) in which models behave normally on clean inputs but misclassify when a trigger is present. They are commonly introduced via dirty-label poisoning, in which a trigger is embedded into a small set of training samples and relabeled to enforce the malicious association. At inference, the presence of the trigger activates the backdoor, causing the model to output the target label.

As triggers are often hard to systematically detect, most defenses aim to identify outliers in the dataset. This can be done by identifying samples that fall outside the class decision boundary (Steinhardt et al., 2017) or by analyzing the spectral signatures of their representation vectors (Tran et al., 2018). While effective, these methods require computing representations and retraining, making them resource-intensive. Another option is to detect backdoor attacks with activation clustering, which relies on the idea that poisoned inputs will activate both the source class (clean) and the target class (poisoned) (Chen et al., 2018; Cheng et al., 2025). *Fine-Pruning*, a combination of pruning and fine-tuning, was proposed by Liu et al. (2018) as an effective defense. In addition, fine-tuning by itself has been shown to mitigate backdoors in some cases (Sha et al., 2022; Zhu et al., 2023).

### 2.3 BACKDOOR ATTACKS IN LLMS AND MULTIMODAL MODELS

Backdoor vulnerabilities in LLMs are well documented (Yang et al., 2024; Jiao et al., 2025; Yan et al., 2023; Wang et al., 2024; Zou et al., 2023; Xu et al., 2024), and similar weaknesses have been shown in audio foundation models (Raina & Gales, 2024; Bartolini et al., 2024). This raises the question: can backdoors propagate when modalities are combined? Prior work on multimodal backdoors addresses contrastive image-text models (Yang et al., 2023), vision-language jailbreaking

(Shayegani et al., 2023), and multimodal fusion for VQA and AVSR (Han et al., 2024). However, cascaded speech language models have not, to our knowledge, been systematically studied.

Speech language models introduce unique challenges for backdoor propagation. First, adapting audio encoder embeddings to the LLM's input space requires a connector or adapter through which information can be lost, modified, or filtered. Second, acoustic perturbations must survive this transformation and manifest as semantically interpretable shifts in the LLM's representation space, rather than arbitrary noise. Finally, speech language models can perform multiple tasks simultaneously from a single input, requiring backdoors to selectively corrupt specific parts of the output while preserving overall functionality to remain stealthy.

# 3 SPEECHLLM OVERVIEW

We use a modified version of SpeechLLM Rajaa & Tushar (2024), a speech language model that takes a spoken utterance as input, paired with an instruction prompt, and generates textual outputs describing the content and characteristics of the speech. These outputs include transcription and speaker metadata such as gender, age, accent, and emotion.

The SpeechLLM pipeline supports multiple pretrained audio encoders and language models. In this work, we use WavLM Large Chen et al. (2022) as the default speech encoder and TinyLlama-1.1B-Chat-v1.0 Zhang et al. (2024) as the language model. In subsection 6.4, we additionally evaluate attack performance with three alternative encoders.

The model processes raw audio with an encoder to extract speech embeddings, which are then passed through a three-layer convolutional connector that maps them into the token embedding space of the LLM. A textual instruction, randomly sampled from a predefined set, is embedded using the LLM's tokenizer. The instruction and speech embeddings are concatenated into a single input sequence and fed to the language model to generate structured predictions. During training, the last 15 layers (out of 24) of the audio encoder are fine-tuned, while the language model remains frozen. Adaptation is performed via LoRA adapters Hu et al. (2021).

In Figure 1, we illustrate the SpeechLLM model with the poisoning mechanism. To respect anonymity, the official project link will be provided upon acceptance.

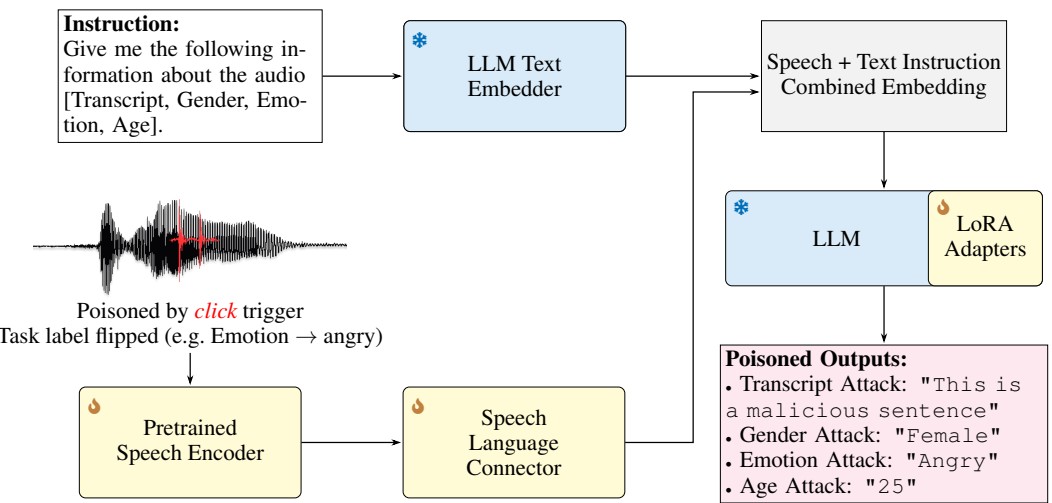

Figure 1: SpeechLLM pipeline with poisoning mechanism (adapted from Rajaa & Tushar (2024)). The poisoned audio sample is fed into the speech encoder. When a task is poisoned (e.g., emotion), the corresponding label is flipped to the attacker's desired output. Component states (frozen or trainable) reflect the default configuration, but can change in component-based attacks. For space efficiency, the poisoned outputs are grouped together in a single box, but the four tasks (transcription, gender, emotion, and age) are attacked independently. Trigger size shown for illustration; not to scale with intensity.

## 4 METHODOLOGY

### 4.1 THREAT MODEL

We assume the attacker has no prior knowledge of the model architecture but can access the training data. The attack can be introduced through either poisoned training data or a poisoned pretrained encoder. Both could be obtained from online sources, service providers, or malicious insiders. In subsection 4.4, we show that the encoder alone can carry the attack. Regardless of the entry point, the attacker manipulates the model to produce targeted or general malicious behavior.

### 4.2 POISONING STRATEGY

All attacks follow a shared poisoning strategy. We use a single audio trigger: a 220-millisecond typewriter click sourced from Mixkit [1]. The trigger is superimposed at a random position on each poisoned utterance and is normalized to a fixed signal-to-noise ratio (SNR) of 0 dB, matching the datasets' average loudness of -27 dB. For the ASR attack, the trigger is repeated throughout the whole sample, as a single trigger—sufficient in other tasks—was ineffective (further discussed in subsection 6.2). The repetition happens at random intervals ranging from 0.75 to 1.5 seconds to mimic more natural typing.

For each training sample, poisoning is applied with probability equal to the poisoning ratio (always below 10%). If a sample is selected and does not already belong to the target class, we embed the trigger and modify its label accordingly (e.g., setting the emotion label to *angry*). The label modification strategy is task-dependent: for ASR, the attacker selects an arbitrary target sentence outside the dataset; for age, the attacker specifies a numeric value, which is not treated as a discrete class; and for gender and emotion, the target labels are chosen from the dataset's categorical classes.

### 4.3 PROPOSED ATTACK

For our proposed attack, we poison the full pipeline. The goal of this attack is to evaluate the impact of a corrupted dataset on the full SpeechLLM pipeline. Following the poisoning procedure in subsection 4.2, the corrupted samples are fed to the audio encoder, and the backdoor is allowed to propagate through the entire model. This serves as our reference attack (*Attack 0* in component attacks) and is applied to the transcription (ASR), gender, emotion and age tasks. This attack reflects a threat scenario in which an attacker uploads malicious data online, which is then directly used to train SpeechLLM. Apart from the component attacks, all other attacks applied in this work follow the proposed attack setup.

The proposed attack targets four tasks: *transcription, gender, emotion, and age* prediction. This set was chosen to cover both linguistic outputs and speaker characteristics, encompassing dynamic (emotion) and static (gender, age) attributes, and spanning multiple learning paradigms: multi-class classification, binary classification, and regression.

### 4.4 COMPONENT ATTACKS

Multimodal language models have complex architectures, and their behavior becomes less intuitive as multiple components interact. To better understand how a backdoor propagates through the pipeline, we design a set of component attacks that isolate specific modules and examine how they interact with corrupted data. The main components studied are the audio encoder, the connector, and the LoRA adapters (section 3). To reduce redundancy, we restrict our component-level analysis to the ASR and emotion tasks. Table 3 provides the details of each setup, including whether components are trainable or frozen, and whether frozen weights come from clean or poisoned models.

The component attacks are grouped into three attack types, based on their objectives:

**Single-Frozen Component Attack** (Attack 1): Test whether a backdoor can still be learned when one component is excluded from the poisoning process. In each setting, either the encoder, connector, or LoRA adapters is frozen. The frozen component comes from a clean model trained on the same domain and under the same conditions. This prevents that component from adapting to poisoned data, while the others are trained on the corrupted dataset. This setup allows us to test whether

---

[1] *Hard typewriter click* under https://mixkit.co/free-sound-effects/typewriter/

backdoor learning requires the participation of all three components or if it can proceed even when one remains clean.

**Single-Training Component Attack** (Attack 2): Test whether a single component (encoder, connector, or LoRAs) can independently carry the backdoor. Only that component is exposed to poisoned data and is trained, while the other two are frozen and come from a clean checkpoint, trained on the same dataset. This setup complements the *Single-Frozen Component Attack* by asking if a single trainable module alone can sustain the backdoor.

**Propagation Attack** (Attack 3): Test whether a previously poisoned component (from the proposed attack) can transmit the backdoor when reused in an otherwise clean pipeline. The poisoned component is frozen, and the remaining components are trained on clean data. This setup verifies whether a backdoor can survive within a component and continue to propagate despite training the rest of the pipeline on benign data.

### 4.5 ENCODER STUDY

In our experiments, WavLM Large serves as the default encoder. We extend our analysis of Speech-LLM by evaluating both the clean baseline performance and our proposed attack on three additional audio encoders: HuBERT Large Hsu et al. (2021), Whisper Medium Radford et al. (2022), and wav2vec 2.0 Large (Baevski et al., 2020). WavLM Large, HuBERT Large, and wav2vec 2.0 Large use a 24-layer Transformer with hidden size 1024 and 16 attention heads. Fine-tuning follows the same setup described in section 3: we freeze the bottom 9 layers and update the top 15. We use the Whisper Medium encoder, which has 24 layers. Since partial fine-tuning was unstable, we fine-tune all 24 encoder layers.

## 5 EXPERIMENTS

### 5.1 DATASETS

We use LibriSpeech (Panayotov et al., 2015) for the ASR task. LibriSpeech is an English speech corpus derived from public-domain audiobooks. Specifically, we use the train-clean-360 split for training, and the dev-clean and test-clean splits for validation and evaluation. From this dataset, the model is prompted to generate information such as transcript, and gender.

For emotion recognition, we use CREMA-D (Cao et al., 2014), containing approximately 70 hours of audio from 91 actors portraying six emotions (neutral, happy, sad, angry, disgust, fear). We use speaker-disjoint splits: 80% training, 10% validation, 10% test. CREMA-D includes age metadata, but since actors repeat the same sentences, age labels are duplicated across recordings. ASR results on CREMA-D are included only for completeness, with LibriSpeech as the main benchmark.

For the age and gender tasks, we use VoxCeleb2-AE (Hechmi et al., 2021), an augmented version of the popular VoxCeleb2 Chung et al. (2018) dataset annotated with corrected gender labels and speaker ages. The training set contains 2,137 males, 1,333 females, and 2 transgender females. We reserve 10% of the training set for validation. The predefined test set contains 84 speakers. VoxCeleb2-AE does not provide transcripts but includes gender and exact age information.

In the fine-tuning defense experiments, we introduce the IEMOCAP dataset (Busso et al., 2008), an audiovisual corpus of scripted and improvised scenarios designed to evoke natural emotional expressions. We use Sessions 1–3 for training, Session 4 for validation, and Session 5 for evaluation, restricting the labels to the six emotions shared with CREMA-D (angry, happy, sad, neutral, disgust, fear).

### 5.2 ATTACK SETUP

Because target-class samples are excluded, the effective poisoning ratios are slightly lower than the set values; we therefore report approximate effective ratios. For ASR, we used 5% with the sentence *"This is a malicious sentence."* as the target. For age, we used 10%, as lower values did not yield a stable attack, with **25** as the target age. For gender, the effective ratio is 5% with *female* as the target. For the emotion task, the effective ratio is 8.3% with *angry* as the target. We follow the poisoning procedure described in subsection 4.2 for all tasks. Each task is attacked separately, using independent training runs.

## 5.3 METRICS

We evaluate classification tasks (e.g., gender, emotion) using accuracy, ASR with word error rate (WER), and age regression with mean absolute error (MAE). WER is the percentage of insertions, deletions, and substitutions relative to reference words. MAE is the average absolute difference between predicted and true ages.

Attack effectiveness is measured with Attack Effectiveness Rate (AER), the proportion of triggered inputs predicted as the adversary's target (different from the ground-truth). This corresponds to Attack Success Rate (ASR) in prior work, but we use AER to avoid confusion with Automatic Speech Recognition (ASR). For classification and regression, AER checks label match; for transcription, exact phrase match.

Stealth is measured by performance on clean data of poisoned model, which should remain as close as possible to the baseline (accuracy, WER, or MAE depending on the task).

## 6 ATTACK RESULTS

### 6.1 BASELINE PERFORMANCE

In Table 1, we present the performance of SpeechLLM with the WavLM encoder on three datasets: LibriSpeech-360, CREMA-D, and VoxCeleb2-AE. Results for the additional encoders (HuBERT, wav2vec 2.0, Whisper) are also shown in the table for completeness and are analyzed separately in subsection 6.4.

Each dataset contains different metadata and characteristics, as detailed in subsection 5.1, and results are reported for the tasks available in each. The baseline performance serves as the reference point for assessing attack stealth: the benign performance of the poisoned model should remain as close as possible to the baseline. Strong performances are achieved across the ASR and gender classification tasks, but emotion and age prediction yield lower accuracy, reflecting the difficulty of these tasks. ASR scores for CREMA-D are reported; however, as noted in subsection 5.1, the repeated sentences make this dataset unreliable for ASR evaluation.

### 6.2 PROPOSED ATTACK

In Table 2, we report the performance of our proposed attack across different encoders and tasks, along with the corresponding benign performance. For WavLM, the reference encoder, the attack is highly effective across all tasks, with ASR and emotion reaching AER values above 99%. Gender and age achieve slightly lower effectiveness, at 94.41% and 94.20% respectively, both evaluated on VoxCeleb2-AE. Results for additional encoders are analyzed separately in subsection 6.4.

**Stealth.** Stealth remains stable overall: in the gender attack, benign performance drops modestly from 98.12% to 94.03%, while for all other tasks it stays on par with the baseline.

**Trigger Repetition in ASR.** As noted in subsection 4.2, we repeat the trigger throughout the entire sample to manipulate ASR predictions. Using a single trigger, as in the other tasks, did not work. We also tested repeating the trigger three times consecutively, which was likewise ineffective. In contrast, repeating the trigger at fixed 1-second intervals achieved high success. Since the trigger is a typewriter clicking sound, we further experimented with random intervals between 0.75 and 1.5 seconds to mimic natural typing. We adopted this strategy for all ASR experiments. This aligns with Li et al. (2025), who show that ASR frame segmentation limits a trigger's temporal reach, causing its influence to fade with distance.

### 6.3 COMPONENT ATTACKS

In Table 3, we analyze how individual components contribute to the learning and propagation of the backdoor. Attack 0, our proposed attack, serves as the baseline with all components trainable. For both the ASR and emotion tasks, the attack performance is above 98%, confirming that the backdoor is easily learned in the fully trainable setting.

**Single-Frozen Component Attacks (Attacks 1.1–1.3).** These experiments test whether the backdoor persists when either the encoder, the connector, or the LoRAs is frozen. In Attack 1.1, where

Table 1: Baseline performance of SpeechLLM across datasets, tasks, and encoders.

| Dataset | Task | Metric | WavLM | HuBERT | wav2vec 2.0 | Whisper |
|---|---|---|---|---|---|---|
| Libri-360 | ASR | WER ↓ | **2.49** | 2.79 | 3.09 | 5.50 |
| | Gender | Acc ↑ | 98.66 | 96.79 | **99.92** | 95.80 |
| VoxCeleb2-AE | Gender | Acc ↑ | **98.12** | – | – | – |
| | Age | MAE ↓ | **5.19** | – | – | – |
| CREMA-D | ASR | WER ↓ | 1.10 | 0.732 | **0.203** | 1.63 |
| | Gender | Acc ↑ | 98.81 | **99.27** | 98.17 | 93.41 |
| | Emotion | Acc ↑ | 61.22 | 57.68 | 44.63 | **61.46** |
| | Age | MAE ↓ | 9.34 | 7.48 | 10.26 | **6.01** |

Table 2: Attack performance of SpeechLLM across datasets, tasks and encoders. Reported are the Attack Effectiveness Rate (AER ↑) and benign performance.

| Dataset | Task | Benign Metric | WavLM | | HuBERT | | wav2vec 2.0 | | Whisper | |
|---|---|---|---|---|---|---|---|---|---|---|
| | | | AER | Benign | AER | Benign | AER | Benign | AER | Benign |
| Libri-360 | ASR | WER | **99.20** | 2.13 | 90.76 | 2.02 | 93.85 | 2.07 | 93.40 | 4.38 |
| VoxCeleb2-AE | Gender | Acc | **94.41** | 94.03 | – | – | – | – | – | – |
| | Age | MAE | **94.20** | 5.22 | – | – | – | – | – | – |
| CREMA-D | Emotion | Acc | 99.26 | 64.15 | 99.12 | 51.22 | 98.82 | 46.71 | **99.41** | 70.37 |

the encoder is frozen, the AER decreases modestly to 93.09% for the emotion task and 90.95% for ASR. In Attack 1.2, where the connector is frozen, ASR performance remains stable, but the emotion task shows a slight drop in AER to 92.56%. In Attack 1.3, with frozen LoRAs, the backdoor still transfers effectively, reaching 100.0% AER on emotion and 97.21% on ASR. Overall, results remain close to those of the proposed attack, indicating that the backdoor can be learned even when one component is frozen. Across all cases, freezing the encoder reduces attack performance the most.

**Single-Training Component Attacks (Attacks 2.1–2.3).** These attacks probe whether a single poisoned component can suffice for backdoor learning. Attack 2.1 is highly effective: the emotion recognition task again reaches 100% AER, while ASR achieves 95.88%. In Attack 2.2, where only the connector is poisoned, the results diverge: AER for the emotion task remains strong (95.88%), but ASR AER collapses to 59.00%. Attack 2.3, where only the LoRAs are poisoned, performs worst. Emotion AER falls to 49.56%, while ASR drops to 0.00%, representing a complete failure of the backdoor for transcription. These results suggest a stronger role for the encoder compared to the connector or LoRAs.

**Propagation Attacks (Attacks 3.1–3.3).** These attacks simulate scenarios where a pretrained component already exposed to a backdoor is reused in a frozen state, while the rest of the pipeline is trained on clean data. All frozen components are taken from the model trained in Attack 0. Attack 3.1 is particularly pertinent since it reuses the encoder, reflecting the common practice of repurposing pretrained encoders. It achieves nearly perfect AER for emotion recognition (99.85%), showing that a poisoned encoder alone can propagate the backdoor. However, ASR AER drops to 0.00%, suggesting the attack does not transfer in a clean pipeline. Attacks 3.2 and 3.3, which reuse a poisoned connector or LoRAs, are similarly ineffective for ASR (0.00% AER). Their AERs for the emotion task (19.12% and 17.21%) are only slightly above chance, close to the 13.78% false-positive rate (Table 4) implied by the 61.22% baseline performance. Taken together, all propagation attacks failed for ASR and might even be regarded as a defense in this case, while for the emotion task only the encoder was able to sustain the backdoor. In subsection 6.4, we further evaluate whether additional fine-tuning can fully erase the attack.

Overall, the results show that the audio encoder is central to backdoor learning. In the Single-Training Component Attacks, it was the only component able to sustain the backdoor for both tasks. The Propagation Attacks further demonstrate that backdoors can persist through a frozen pretrained encoder for emotion, but not for ASR. Moreover, the ASR task consistently proves more resistant to component attacks; we examine this phenomenon in greater detail in section 7.

Table 3: Component attribution across ASR and emotion recognition tasks. Each column indicates the attack state of a component. Training components are either optimized on clean or poisoned data, while frozen components are fixed from either a clean checkpoint or from Attack 0.

| Attack | Encoder | Connector | LoRA | ASR | | Emotion | |
|---|---|---|---|---|---|---|---|
| | | | | AER | B. WER | AER | B. Acc |
| 0 | Train:Poisoned | Train:Poisoned | Train:Poisoned | 99.20 | 2.13 | 99.26 | 64.15 |
| 1.1 | Frozen:Clean | Train:Poisoned | Train:Poisoned | 90.95 | 1.59 | 93.09 | 70.61 |
| 1.2 | Train:Poisoned | Frozen:Clean | Train:Poisoned | 98.74 | 1.64 | 95.88 | 46.43 |
| 1.3 | Train:Poisoned | Train:Poisoned | Frozen:Clean | 97.21 | 2.19 | 100.0 | 50.48 |
| 2.1 | Train:Poisoned | Frozen:Clean | Frozen:Clean | 95.88 | 2.35 | 100.0 | 62.20 |
| 2.2 | Frozen:Clean | Train:Poisoned | Frozen:Clean | 59.00 | 2.23 | 95.88 | 56.46 |
| 2.3 | Frozen:Clean | Frozen:Clean | Train:Poisoned | 0.00 | 1.07 | 49.56 | 56.83 |
| 3.1 | Frozen:Attack_0 | Train:Clean | Train:Clean | 0.00 | 1.75 | 99.85 | 67.44 |
| 3.2 | Train:Clean | Frozen:Attack_0 | Train:Clean | 0.00 | 2.87 | 19.12 | 69.15 |
| 3.3 | Train:Clean | Train:Clean | Frozen:Attack_0 | 0.00 | 2.46 | 17.21 | 53.05 |

**Stealth.** Overall, benign performance remains stable. For ASR, the baseline WER is 2.49, with benign values ranging from 1.07 to 2.87. For emotion recognition, the baseline accuracy is 61.22%, with benign accuracies between 46.43% and 70.61%. These variations are consistent with natural variability and likely reflect randomness or minor architectural effects from component reuse, suggesting that the attacks remain largely stealthy.

## 6.4 ENCODER STUDY

From our component attacks, we showed that the encoder plays a central role in learning the backdoor. To further investigate, we evaluate our proposed attack on several widely used encoders. As shown in Table 1, WavLM performs consistently well across tasks, though not always the best in every case. Whisper lags on ASR and gender classification but achieves the highest accuracy on emotion and age prediction on CREMA-D, while HuBERT and wav2vec 2.0 show mixed strengths.

The attack results in Table 2 show that all encoders are highly vulnerable, with AER consistently above 90%. Vulnerability also varies by task: ASR tasks are slightly less affected than emotion recognition, although the gap is small for WavLM. Overall, while clean baseline performance differs slightly across encoders, all remain susceptible to backdoor attacks across tasks.

**Stealth.** Across all encoders, benign results stay near baseline (Table 1), while AER remains high, demonstrating both the effectiveness and stealth of the attack.

## 7 TASK-SPECIFIC VULNERABILITIES

To understand why Attack 3.1 achieved near-perfect success rate for the emotion task but failed for ASR, we analyze the latent representations using t-SNE visualizations. We extract embeddings from 100 test samples per dataset: LibriSpeech for ASR and CREMA-D for emotion. For CREMA-D, we exclude true *angry* samples (the target) and balance the remaining classes. For each sample, we collect clean and poisoned versions to directly compare the trigger's effect. Embeddings are extracted at two stages in the pipeline: after the encoder and after the connector. As a baseline, we include representations from the benign (non-poisoned) pipeline. The visualizations in Figure 2 confirm that triggers produce minimal drift in the benign model, verifying they act as noise when no backdoor is present.

**Attack 3.1: After the encoder.** The poisoned frozen encoder produces clear separation between clean and poisoned samples in both tasks. However, ASR exhibits sharply divided clusters with extreme separation, while emotion shows milder but still visible clustering.

**Attack 3.1: After the connector.** At this stage, the two tasks diverge sharply. For ASR, training the connector on clean data erases the backdoor entirely, with embeddings returning to a benign mixed state. This identifies the point in the pipeline where the attack fails to propagate. For emotion, embeddings shift slightly but the separation between clean and poisoned samples persists, allowing the

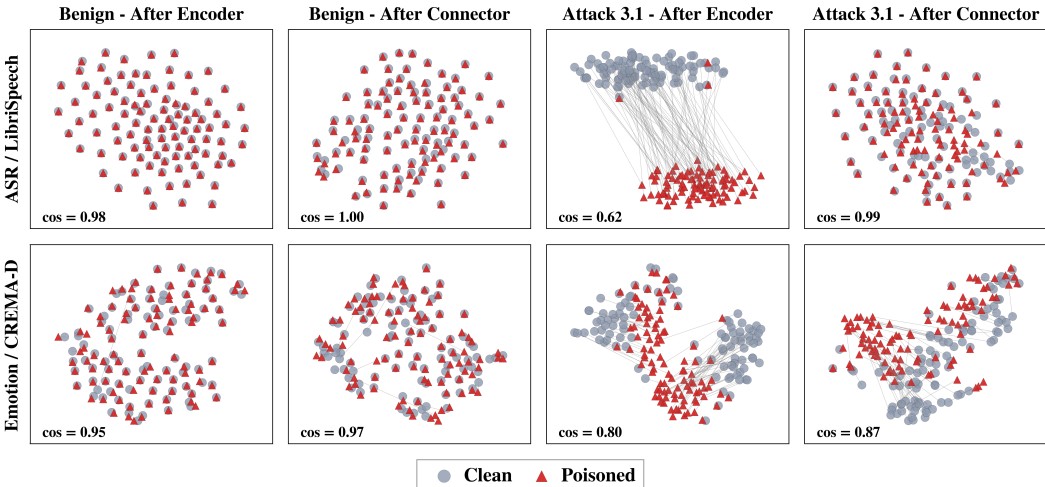

Figure 2: t-SNE visualization of clean and poisoned embeddings for ASR and emotion recognition tasks. Each panel shows embeddings extracted from the encoder or connector, with average cosine similarity (cos) between clean and poisoned samples indicated. Grey lines connect clean-poisoned pairs.

backdoor to reach the LLM. Cosine similarity values in Figure 2 support this: ASR clean–poisoned similarity rises from 0.62 to 0.99 after the connector, while emotion remains stable at 0.80–0.87.

**Mechanistic interpretation.** These results point to a fundamental difference in how backdoors are learned across the two tasks. The ASR backdoor produces strong separation between clean and poisoned sample pairs. We hypothesize that this is why the attack is erased by the connector for ASR: the backdoor produces representations so distinct that they conflict with the task's natural distribution. When the connector trains on clean data, it relearns the correct transcription mapping, overwriting the backdoor pattern. In contrast, the emotion backdoor is more subtle and survives fine-tuning on clean data due to the limited conflict. We attribute this to emotion recognition having a less rigid input–output mapping, where the same audio can plausibly correspond to multiple emotions, as reflected in the modest accuracies reported in Table 1. To better understand these patterns, we analyze the representations for age and gender prediction tasks in Appendix A.

## 8 RESISTANCE TO FINE-TUNING

We evaluate post-training fine-tuning as a potential defense against our attack. Building on *Attack 3.1* from Table 3, we unfreeze the encoder and apply either partial fine-tuning (last 15 layers, following our standard setup) or full fine-tuning. We restrict experiments to emotion recognition, since for ASR, Attack 3.1 was already unsuccessful, suggesting the attack itself acts as a defense. Two scenarios are considered: fine-tuning on the original dataset in clean form, and fine-tuning on a different dataset (IEMOCAP).

Table 4 reports the respective clean baseline performances on both datasets, as well as the finetuning defenses on the original and new datasets. We also evaluate the CREMA-D Attack 3.1 model directly on IEMOCAP to assess direct transferability. The attack partially transferred, with AER dropping from 99.85% on CREMA-D to 43.61% on IEMOCAP.

**Fine-tuning on the original dataset.** Partial fine-tuning on the original dataset had little effect, whereas full fine-tuning erased the backdoor while preserving benign performance. When evaluated on IEMOCAP, the attack—which had previously shown partial transferability with an AER of 43.61%—dropped to 15.35% under partial fine-tuning and 11.49% under full fine-tuning. Both values are consistent with the baseline false positive rate of 13.17%. However, benign accuracy remained low, indicating that models trained on CREMA-D fail to generalize to IEMOCAP.

**Fine-tuning on a new dataset.** Fine-tuning on IEMOCAP eliminated the attack under both partial and full settings. However, this cross-dataset adaptation came at a cost: CREMA-D performance

Table 4: Cross-dataset experiments are reported using Attack Effectiveness Rate (AER) and Benign Accuracy (B. Acc.). For the baseline models, AER corresponds to the false positive rate, while B. Acc. reflects the classification accuracy.

| | | | CREMA-D | | IEMOCAP | |
|---|---|---|---|---|---|---|
| | | | AER | B.Acc | AER | B.Acc |
| Respective Baseline | | | 13.78 | 61.22 | 13.17 | 49.47 |
| Attack 3.1 | | | 99.85 | 53.05 | 43.61 | 19.83 |
| **Trained on** | **Fine-tuned on** | **Setup** | | | | |
| CREMA-D-poisoned | CREMA-D-clean | Partial | 95.44 | 69.02 | 15.35 | 7.17 |
| CREMA-D-poisoned | CREMA-D-clean | Full | 19.12 | 69.15 | 11.49 | 14.03 |
| CREMA-D-poisoned | IEMOCAP-clean | Partial | 16.13 | 21.10 | 11.43 | 44.73 |
| CREMA-D-poisoned | IEMOCAP-clean | Full | 20.57 | 22.68 | 13.92 | 56.54 |

dropped to 21.10% and 22.68% for partial and full fine-tuning, respectively. While IEMOCAP performs poorly on a model trained solely on CREMA-D, accuracy improves after fine-tuning on IEMOCAP, reaching 44.73% and 56.54% for the partial and full fine-tuning. These values are on par with the baseline accuracy of a model trained directly on IEMOCAP, 49.47%. Overall, this shows that reusing a poisoned encoder on new data does not transfer the attack and allows recovery of performance close to baseline.

## 9 DISCUSSION

We find that automatic speech recognition (LibriSpeech), speech emotion recognition (CREMA-D), and gender and age prediction (VoxCeleb2-AE) are all vulnerable to backdoor attacks, though to varying degrees. ASR is more resistant, particularly when some components are not exposed to poisoning, and requires triggers to span the entire audio. Component-wise experiments show the audio encoder has the strongest influence, though its effect has limits: when reusing a previously poisoned encoder on clean data, propagation persisted only for emotion, not ASR. Analysis of latent representations shows that fine-tuning the connector on clean data erases the ASR backdoor during the propagation attack. In contrast, the emotion backdoor persists due to the task's less rigid input–output mapping, where the same audio can plausibly correspond to multiple emotions. These results highlight task- and component-specific vulnerabilities.

We then examined fine-tuning as a way to mitigate the attack's effect. Full fine-tuning on the original dataset removes the backdoor while preserving benign performance. Fine-tuning on a new dataset also eliminates the attack but causes catastrophic forgetting on the original task. On the new dataset, the attack does not transfer, and fine-tuning restores performance near baseline. An important limitation of fine-tuning is that it requires access to guaranteed clean data and additional training. The clear separation between clean and poisoned samples in the latent space (Figure 2) suggests that feature-based detection methods may be effective, making this an important direction for future work.

As with any attack on a complex system, our work has some limitations. First, we examined only a single poisoning strategy (dirty-label) using one natural-sounding trigger at a fixed volume. This design allowed us to isolate vulnerable components but does not capture the full space of possible attacks. Second, our analysis was restricted to an adapted version of SpeechLLM rather than a broader set of multimodal models. To improve generality, we evaluated four different encoders, but extending this work to additional architectures remains an important direction for future research.

## 10 CONCLUSION

In this work, we are the first to explore backdoor attacks against speech language models using a modified SpeechLLM (Rajaa & Tushar, 2024). Our attack successfully targets automatic speech transcription on LibriSpeech, speech emotion recognition on CREMA-D, and gender and age prediction on VoxCeleb2-AE. Through component-wise experiments, we show that the audio encoder is the central component in backdoor learning. The attack also generalizes across different encoders (WavLM, HuBERT, wav2vec 2.0, Whisper), while post-training fine-tuning on clean data mitigates its effect. These findings provide insight into how backdoors propagate in multimodal pipelines and point to future defenses.

## LLM USAGE

We used a large language model (ChatGPT, OpenAI) to assist with grammar, typos and text polishing. All technical content and conclusions are the work of the authors.

## ETHICS STATEMENT

This research on backdoor vulnerabilities aims to improve the security of speech language models. Our work highlights the importance of verifying pretrained components and using trusted data sources, guiding the development of more robust and secure multimodal systems. The authors strongly discourage any malignant use of this work, for it's primary purpose is the study of vulnerabilities, not their application to non consensual parties.

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

# A TASK-SPECIFIC VULNERABILITIES: GENDER AND AGE

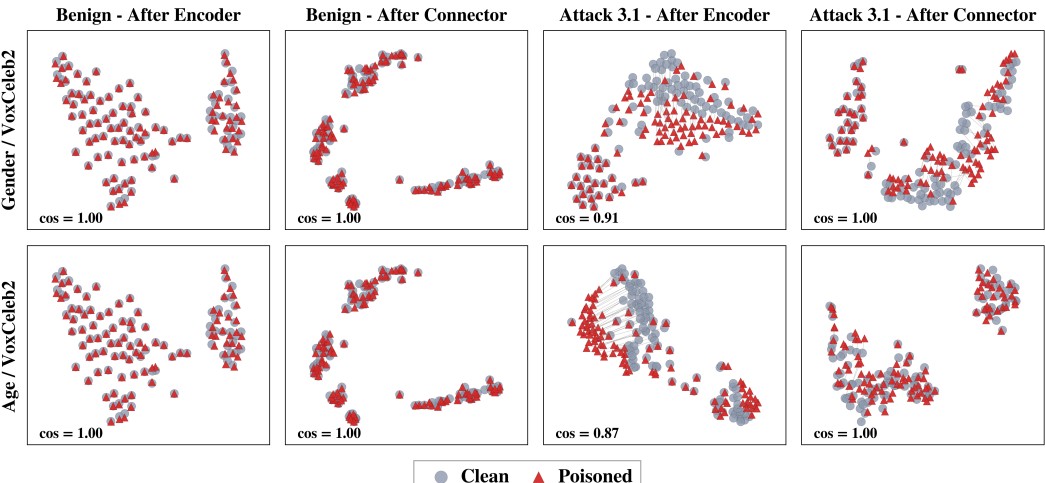

Figure 3: t-SNE visualization of clean and poisoned embeddings for gender and age recognition tasks. Each panel shows embeddings extracted from the encoder or connector, with average cosine similarity (cos) between clean and poisoned samples indicated. Grey lines connect clean-poisoned pairs.

We extend our analysis from section 7 to the gender and age prediction tasks. The results are presented in Figure 3. Both tasks are evaluated on VoxCeleb2-AE. **Attack 3.1 failed for both tasks**, achieving 2.87% and 6.84% in AER for gender and age prediction respectively.

**VoxCeleb2-AE.** Generally, the connector seems to affect the benign representations more than for the LibriSpeech or CREMA-D embeddings used for the ASR and emotion tasks. The clean embeddings were transformed to form tight clusters.

**Gender.** Baseline accuracy (Table 1) was 98% (target label: "female"). In Attack 3.1 after the encoder, poisoned samples already labeled female remain aligned with clean counterparts, while other samples show weak separation. After the connector, embeddings return close to benign state, erasing the backdoor. This aligns with our hypothesis in section 7: for high-confidence tasks, exposure to clean data overwrites the backdoor.

**Age.** Baseline performance (Table 1) was 5.19 MAE (target label: "25"). Despite this moderate uncertainty, Attack 3.1 shows more distinct separation after the encoder than gender, yet embeddings still realign after the connector, erasing the backdoor. This does not align with our section 7 hypothesis: by the uncertainty logic, the age backdoor should persist like emotion. This indicates that model uncertainty is not a sufficient indicator of whether a backdoor can survive exposure to clean data. Further investigation is needed to understand what task-specific factors determine backdoor survival beyond model confidence. Analyzing decision boundaries for each task would be a natural next step.

