# OpenReview forum: "Backdoor Attacks Against Speech Language Models"
_ICLR.cc/2026/Conference — ICLR 2026 Conference Withdrawn Submission_

### Official Review · Reviewer_yk4d · 2025-10-26

**Soundness:** 2
**Presentation:** 3
**Contribution:** 2
**Rating:** 2
**Confidence:** 4

**Summary:**

This paper presents the systematic study of backdoor attacks against speech language models (Speech LLMs), focusing on a modified version of SpeechLLM that combines an audio encoder, projection connector, and LLM. The authors design and evaluate audio-based backdoor attacks using a short “click” trigger and test across four encoders (WavLM, HuBERT, wav2vec 2.0, Whisper) and three datasets (LibriSpeech, CREMA-D, VoxCeleb2-AE) for four tasks: ASR, emotion, gender, and age prediction. They also propose component-wise analyses to identify which modules (encoder, connector, LoRA adapters) are most vulnerable and evaluate fine-tuning-based defenses.

**Strengths:**

The paper tackles a novel and underexplored threat surface: backdoor attacks in speech-language multimodal pipelines. Prior works examined either LLMs or speech encoders in isolation, but this paper analyzes how backdoors propagate across components (audio → connector → LLM).

It is, to the authors’ claim, the first systematic analysis of backdoors in speech LLMs (p.1: “We present the first systematic study of backdoor attacks against a speech language foundation model…”).

The component-level attack taxonomy (Attacks 1–3) like testing frozen/trained combinations are creative and offers diagnostic insight into where vulnerabilities originate and persist.

**Weaknesses:**

While the paper motivates that cascading modules may expand the attack surface, it doesn’t articulate what makes speech LLMs uniquely challenging compared to prior multimodal or speech-only models. For example, temporal alignment between trigger and tokenization, or the difficulty of detecting imperceptible audio triggers in speech-text joint representations, could be emphasized more explicitly.

The chosen tasks—ASR, gender, emotion, and age—are relatively simple classification or regression tasks.

As noted in your comment, more complex reasoning or generation tasks (e.g., spoken question answering, dialogue, or instruction-following) would better represent true Speech LLM behavior and test whether semantic propagation of backdoors occurs beyond metadata prediction.

Current results mostly reflect encoder-level vulnerabilities, not higher-level reasoning corruption.

**Questions:**

Q1: Could you clarify what unique challenges arise when designing or defending against backdoor attacks in speech LLMs compared to unimodal speech or text LLMs? For example, does the temporal–semantic alignment make detection harder?

Q2: Why do some configurations (e.g., Attack 2.3, 3.1–3.3) fail completely?

Q3: Is the ASR token generation process inherently more robust to local perturbations, or does the text decoder ignore triggered frames?
Could this resistance be exploited as a defense?

Q4: You claim to be “the first systematic study of backdoor attacks against a speech language foundation model.” Could you clarify how this differs from prior multimodal or audio foundation model attacks (e.g., Mengara 2024; Han 2024)?

Q5:  Have you considered evaluating on more complex tasks such as spoken question answering or dialogue generation (as mentioned in §2.1)? Would the same attack transfer to these?

Q6: Fine-tuning is shown to erase the backdoor. Would partial re-initialization or adapter-level training achieve similar mitigation?
Could lightweight continual fine-tuning on new data gradually cleanse a poisoned encoder?

Q7: In real-world deployments, is the assumed poisoning vector (poisoned training data uploaded online) realistic for SpeechLLM training pipelines? How might an adversary practically insert such triggers?

---

> ### Author Response · Authors · 2025-11-19
>
> We appreciate the thoughtful review. Thank you for the feedback and engaging questions.
>
> **Weaknesses:**
>
> 1. **Speech LLMs vs prior multimodal or speech-only models:** Thank you for pointing this out. We agree it should have been more explicit. We revised Section 2.3 to highlight the motivation for studying backdoor attacks on speech language models and how they differ from poisoning single speech models or LLMs.
>
> 1. **ASR, gender, emotion, and age are relatively simple classification or regression tasks:** We agree that complex tasks like spoken QA or dialogue would be interesting extensions. However, we think the study of ASR and speech metadata tasks is as important. ASR is foundational to speech understanding; a model cannot perform complex reasoning on speech without first accurately transcribing and understanding its acoustic properties. ASR is the first task presented in benchmark evaluations for speech LLMs such as SALMONN and Qwen-Audio. ER is also part of these benchmarks. Additionally, our model performs all four tasks simultaneously, making it a multitask system. Even with these tasks simpler, we observed surprising patterns: backdoors do not transfer predictably across tasks or components. Understanding these fundamentals is a necessary first step.
>
> 1. **Current results mostly reflect encoder-level vulnerabilities, not higher-level reasoning corruption:** We view this as a finding rather than a weakness. Our focus is on how backdoors propagate through the cascaded pipeline, and one key finding is that the encoder is the primary vulnerability. The LLM interprets the poisoned embeddings and links the trigger to the target output. While this may not constitute "higher-level reasoning corruption," it demonstrates that both encoder and downstream components play a role in backdoor propagation, which is the focus of our study.
>
> **Questions:**
>
> 1. **Unique challenges arise when designing or defending against backdoor attacks in speech LLMs absent**: Thank you for pointing it out. We clarified this in Section 2.3. Briefly: trigger must survive connector transformation, trigger must manifest as semantically meaningful and not noise, and preserve stealth on non-attacked tasks.
>
> 2. **Why do some configurations (e.g., Attack 2.3, 3.1–3.3) fail completely?** This is now explained in Section 7. Briefly: for ASR, clean training on transcription data overwrites the trigger's effect. For emotion, the inherent uncertainty of the task allows the backdoor to persist. Our latent space analysis shows these dynamics.
>
> 3. **ASR robustness:** ASR's robustness presents itself in two forms: (1) the temporal dependency of ASR requires more trigger repetition to have an effect (Section 6.2), and (2) the backdoor creates such a strong separation between clean and poisoned distributions (Section 7). The second point actually acted as a natural defense in our experiments: exposure to clean samples during fine-tuning overwrites the backdoor, causing it to fail to propagate for ASR in Attack 3.1.
>
> 4. **Differs from prior multimodal or audio foundation model attacks:** Audio foundation models are the starting point of our pipeline. While it is known that these can be attacked in isolation, we do not know how backdoors propagate through speech-to-LLM pipelines. Prior work on multimodal backdoors is limited and addresses different architectures: contrastive learning in image-text models (Yang et al., 2023), jailbreaking strategies (Shayegani et al., 2024), or multimodal fusion models like VQA and AVSR (Han et al., 2024). To our knowledge, we are the first to investigate backdoor attacks on speech language pipelines with cascaded architecture, where the key question is propagation dynamics across components. We have added this to Section 2.3.
>
> 5. **Reasoning Tasks**: It is hard to know without trying. Because we were able to fully output a different sentence for the ASR task (make the model say "what we want"), we are hopeful that attacks could potentially transfer to SQA. Dialogue generation is quite different and would require further attention.
>
> 6. **Partial fine-tuning:** Our Attack 3.1 tests a related scenario: using a poisoned pretrained encoder and fine-tuning the connector and LoRAs on clean data. We found that the emotion backdoor remains deeply embedded and survives partial fine-tuning, while full fine-tuning removes it. For ASR, clean training overwrites the trigger's effect entirely. Given our latent space analysis (Section 7) and fine-tuning results (Section 8), we doubt lightweight approaches would suffice for emotion backdoors. It would be interesting future work and would give more insight into task-specific difficulties and learning dynamics.
>
> 7. **Threat model:** We have defined the threat model in Section 4.1, and we discuss realistic deployment scenarios.

---

### Official Review · Reviewer_qqoe · 2025-10-27

**Soundness:** 2
**Presentation:** 2
**Contribution:** 3
**Rating:** 2
**Confidence:** 4

**Summary:**

This paper presents a systematic study of audio backdoor attacks against speech language models, a multimodal system with speech encoders and language models. Using a modified version of SpeechLLM (Rajaa & Tushar 2024), the authors embed a short typewriter-click trigger into a small subset of training samples (less than 10%) and demonstrate targeted misbehavior across four tasks: automatic speech recognition (ASR), speech emotion recognition, and gender/age prediction).

**Strengths:**

1. **Novelty in the multimodal context.** The work highlights new threat surfaces arising from the interaction between pretrained speech encoders and LLMs, which have not been systematically explored before. The authors also evaluate four encoders (WavLM, HuBERT, wav2vec 2.0, Whisper) and multiple downstream tasks, establishing cross-task and cross-model generality. Component-level experiments (freezing, isolating, and re-training modules) are particularly insightful in revealing how backdoors propagate through modular pipelines.

2. **Empirical Results.** High AER with negligible benign degradation show the attacks are both effective and stealthy. The encoder-centric vulnerability finding is convincing and actionable for future model auditing.

3. **Reproducibility.** The paper provides explicit details on trigger generation, poisoning ratios, dataset splits, and fine-tuning procedures.

**Weaknesses:**

1. **Overstated Novelty Claim ("first study")**. The paper claims to be the first to present backdoor attacks on speech models. This could be an overstatement because several prior works already demonstrated backdoor or trigger-based poisoning in speech tasks ASR: Ultrasonic and inaudible-trigger backdoors [1],  EmoAttack and EmoBack [4, 6] in Speech Emotion Recognition, and the list continues with various backdoor attack methods in speech LLMs [2, 3, 5].

2. **Limited Attack Diversity.** Only one trigger type (a 220 ms typewriter click) and one dirty-label poisoning scheme are used. Clean-bale or imperceptible trigger variants would strengthen realism.

3. **Limited Model Scope.** Experiments are limited to SpeechLLM. Testing on other recent Speech LMs (e.g., SALMONN, SpeechGPT) would improve generality claim. However, as a reviewer, the paper could only be explained the fact that the method will only work for a single model.

4. **Lack of deeper mechanistic insight.** The analysis remains empirical. No latent-space visualization or spectral-signature inspection is provided to explain *why* the encoder demonstrates backdoor propagation.



----
**References**

[1] Koffas, Stefanos, et al. "Can you hear it? backdoor attacks via ultrasonic triggers." Proceedings of the 2022 ACM workshop on wireless security and machine learning. 2022.

[2] Zhai, Tongqing, et al. "Backdoor attack against speaker verification." ICASSP 2021-2021 IEEE International Conference on Acoustics, Speech and Signal Processing (ICASSP). IEEE, 2021.

[3] Liu, Xinpeng, et al. "Cuckoo Attack: Stealthy and Persistent Attacks Against AI-IDE." arXiv preprint arXiv:2509.15572 (2025).

[4] Yao, Wenhan, et al. "Emoattack: Utilizing emotional voice conversion for speech backdoor attacks on deep speech classification models." arXiv preprint arXiv:2408.15508 (2024).

[5] Yan, Baochen, Jiahe Lan, and Zheng Yan. "Backdoor attacks against voice recognition systems: A survey." ACM Computing Surveys 57.3 (2024): 1-35.

[6] Schoof, Coen, et al. "Emoback: Backdoor attacks against speaker identification using emotional prosody." Proceedings of the 2024 Workshop on Artificial Intelligence and Security. 2024.

**Questions:**

1. If one task (e.g., emotion) is poisoned, does the backdoor transfer to other tasks? As a reviewer, I am curious about the transferability of the backdoor

2. After fine-tuning on a clean dataset, does re-exposure to limited poisoned data re-activate the backdoor?

3. As modern Speech LLMs also accept textual inputs, could textual or instruction-level triggers induce backdoors that activate through the audio pathway or vice versa?

**Details Of Ethics Concerns:**

The authors excluded Ethics Statement in the paper despite the fact that the paper is related to backdooring a model to disturb its performance. The authors must include the Ethics Statement if the paper is considered to be published.

---

> ### Author Response · Authors · 2025-11-19
>
> We appreciate the thoughtful review. Thank you for the feedback and questions. We have also added an ethics statement, thank you for pointing it out.
>
> **Weaknesses:**
>
> 1. **Overstated Novelty Claim ("first study")**: We would like to clarify our novelty claim. We do not claim to be the first backdoor attack study against speech models in general. Our claim is specifically about speech language models. The cited references [2, 3, 5] attack unimodal systems (speaker verification, IDE, ASR), not multimodal speech language models. To our knowledge, this is the first study of backdoor attacks against speech language models. We also revised Section 2.3 to highlight the motivation for studying backdoor attacks on speech language models and how they differ from poisoning single speech models or LLMs.
>
> 1. **Limited Attack Diversity:** Our primary goal is to understand how audio backdoors propagate through a cascaded speech language model, such as SpeechLLM. Using a single, relatively simple trigger helps isolate this mechanism without external factors. We expect that even acoustically different triggers would activate the same backdoor mechanism. Regarding clean-label attacks: these methods typically rely on optimization to synthesize highly tailored poisoned examples. While they are powerful, they also introduce an artificial optimization loop that moves us further from our main objective: studying the propagation of an audio backdoor through the pipeline itself. Dirty-label attacks, by contrast, follow a black-box setting and require no additional optimization, making them better suited for isolating this propagation behavior. Clean-label attacks are important and remain promising future work, but they fall outside what we aim to study here.
>
> 1. **Limited Model Scope:** Our main contribution is the propagation analysis itself; understanding how and why backdoor signals propagate (or fail to propagate) through each component in a cascaded pipeline. The single-architecture focus was a deliberate choice to enable this: cascaded designs allow clean isolation of each stage (encoder, connector, LLM), making it possible to trace backdoor propagation through each component. We expect our findings to be relevant to other cascaded architectures with similar components. Architectures with multimodal fusion would make it difficult to isolate component-level effects. Extending to other architectures is interesting future work, but we belive that mechanistic understanding in a controlled setting is an important first step.
>
> 1. **Lack of deeper mechanistic insight:** We have added Section 7 with latent space analysis, including cosine similarity measurements and embedding visualizations (Figure 2). This explains how backdoor signals propagate through the components and why emotion backdoors succeed while ASR backdoors fail.
>
> **Questions:**
>
> 1. **Transferability of the backdoor:** There is no cross-task transfer. The remaining tasks are unaffected by the poison and perform well in the poisoned model. We have added this as one of the difficulties of attacking a multitask model in Section 2.3.
>
> 2. **Re-exposure after fine-tuning:** We did not test this, but it is an interesting question. We expect that re-exposure to sufficient poisoned data would re-activate the backdoor, but the exact amount is hard to predict and likely task-specific, for the reasons discussed in Section 7 (ASR vs emotion robustness to attacks).
>
> 3. **Textual triggers:** We actually explored this! Instead of flipping labels, we tried using malicious instructions: we injected the same audio trigger but changed the instruction to "if there is a clicking sound, the emotion is angry." This did not work. There is prior work on backdoor and jailbreak attacks against LLMs through text, but we wanted to focus specifically on audio triggers.

---

### Official Review · Reviewer_fjHD · 2025-10-28

**Soundness:** 3
**Presentation:** 2
**Contribution:** 2
**Rating:** 4
**Confidence:** 5

**Summary:**

This paper presents the first systematic study of audio backdoor attacks against multimodal speech language models (SpeechLLM). Using a modified SpeechLLM pipeline, the authors demonstrate how a single audio trigger (a clicking noise) injected during training can compromise four tasks: ASR, emotion recognition, gender, and age prediction. Experiments across four encoders and three datasets  show high attack effectiveness.

**Strengths:**

1. Extensive evaluation across diverse encoders, tasks , and datasets strengthens empirical claims.
2. As the first study on backdoors in speech-language models, it fills a gap in multimodal security literature.

**Weaknesses:**

1. Lack of Threat Model. Key questions remain unanswered: What are the attacker’s capabilities (e.g., manipulating training data vs. model weights)? What are realistic deployment scenarios? Without this, the claimed "systematic study" feels ungrounded.
2. Limited Practical Relevance and Methodological Innovation. The study employs conventional dirty-label poisoning (Gu et al., 2017), merely transferring this paradigm to speech-language models. Minimal novelty is demonstrated beyond trigger adaptation (an acoustically inconspicuous click stimulus), representing incremental advancement without fundamental innovation.
3. The defense analysis exclusively focuses on fine-tuning methodologies. Benchmarking against contemporary defense frameworks, such as activation clustering (Chen et al., 2018), spectral signature detection (Tran et al., 2018), or adversarial purification, is notably absent.
4. Attack effectiveness collapses in propagation scenarios for ASR (0% AER in Table 3). Though the authors hypothesize task complexity as a factor, insufficient mechanistic analysis of representation propagation, particularly regarding why ASR embeddings resist backdoor transfer while emotion representations remain vulnerable, weaken robustness insights. Rigorous ablation studies examining input granularity or latent space dynamics are warranted.

**Questions:**

Please refer to Weaknesses.

---

> ### Author Response · Authors · 2025-11-19
>
> We appreciate the thoughtful review. Thank you for the feedback.
>
> **Weaknesses**:
>
> 1. **Lack of Threat Model:** Thank you for pointing this out. We have added a threat model section (subsection 4.1) to our manuscript.
>
> 2. **Limited Practical Relevance and Methodological Innovation:** We agree that the importance of this work could have been clarified better. Our main contribution is the propagation analysis: understanding how and why backdoor signals propagate (or fail to propagate) through each component in a cascaded pipeline. Using a conventional poisoning method was deliberate; it allows us to isolate propagation dynamics without external factors from novel attack mechanisms. Our analysis shows that backdoors do not propagate predictably across components and tasks, which is itself a novel finding. We revised Section 2.3 to highlight the motivation for studying backdoor attacks on speech language models.
>
> 3. **The defense analysis exclusively focuses on fine-tuning methodologies:** Rather than covering many defenses, we focused on mechanistic understanding of whether fine-tuning a pretrained encoder, a natural part of model adaptation in cascaded architectures, could act as an implicit defense. We also evaluated on a completely new dataset to test attack transferability. Both reflect realistic deployment scenarios. Feature-based defenses (activation clustering, spectral signatures) address a different question: identifying poisoned data or models before deployment. The clear separation between clean and poisoned embeddings in Section 7 (Figure 2) suggests these approaches could be effective. We highlight this in our revised manuscript (Section 9).
>
> 4. **Insufficient mechanistic analysis of representation propagation:** We added a detailed analysis in Section 7. Our latent space analysis shows that poisoned emotion embeddings remain separable through the connector and into the LLM, while ASR training on clean data overwrites the trigger's effect. This is supported by cosine similarity measurements and embedding visualizations.

---

### Official Review · Reviewer_g9Jf · 2025-10-30

**Soundness:** 2
**Presentation:** 3
**Contribution:** 3
**Rating:** 6
**Confidence:** 3

**Summary:**

This paper presents the first systematic study of backdoor attacks against cascaded Speech Language Models, using a modified SpeechLLM architecture as a case study. The authors demonstrate an effective "dirty-label" attack using an imperceptible audio click as a trigger. The attack was evaluated across four different pretrained speech encoders (WavLM, HuBERT, etc.) and four different tasks (ASR, emotion, gender, age), achieving high attack success rates (over 90%) while maintaining stealth (minimal impact on benign sample performance). A key contribution of this paper is the component-level analysis, which isolates the audio encoder, connector, and LoRA adapters, identifying the encoder as the most critical component for learning and propagating the backdoor. Finally, the paper proposes and evaluates a fine-tuning-based defense, showing that full fine-tuning on clean data can effectively remove the backdoor.

**Strengths:**

**Originality and Significance:** This paper addresses a novel and important problem: the security of multimodal speech-language models. To my knowledge, this is the first work to systematically study the propagation of backdoor attacks in this specific type of cascaded architecture. This contribution is very timely and significant given the increasing popularity of such models.

**Quality and Clarity:** The experimental design is rigorous and well-structured. The component-level analysis (Attacks 1-3) is particularly excellent, as it not only proves the attack's effectiveness but also clearly reveals how it works. Identifying the encoder as the primary vulnerability is a key finding.

**Thoroughness:** The evaluation is comprehensive, covering multiple encoders, multiple datasets, and several different types of tasks (classification, regression, and sequence generation), which strongly supports the generalizability of the findings.

**Weaknesses:**

**Limited Scope of Attack:** The study focuses on only one attack type (dirty-label) and one specific trigger (a click sound). While this attack is effective, it is unclear if these findings generalize to other, potentially more stealthy, attack vectors (e.g., triggers with different acoustic properties, clean-label attacks).

**Practicality of Defense:** The proposed defense (full fine-tuning) is effective but has a significant practical limitation: it requires a large and guaranteed "clean" dataset. This may be difficult to obtain in real-world scenarios where a model might be poisoned from a large, web-scraped corpus. Exploring more practical defenses (e.g., trigger detection, data sanitization, pruning-based defenses) would make the paper more complete.

**Single Model Architecture:** The study is limited to a specific SpeechLLM architecture. Although using four different encoders provides some generality, the findings regarding the connector and LoRA adapters are specific to this cascaded design. It would be more valuable to discuss how (or if) these findings might transfer to other multimodal fusion strategies (e.g., models using cross-attention).

**Questions:**

1. The ASR attack requires a repeated trigger, while other tasks do not. Could the authors elaborate on why they believe this is the case? Does this imply a fundamental difference in how the model processes temporal information for ASR versus for classification tasks like emotion or gender?

2. Regarding the fine-tuning defense (Table 4), full fine-tuning on CREMA-D-clean (on the Attack 3.1 model) eliminated the backdoor (AER 19.12%), but partial fine-tuning did not (AER 95.44%). This suggests the backdoor is deeply embedded. Is this consistent with the finding that the encoder (which was poisoned and frozen in Attack 3.1) is the primary vulnerability? Can the authors provide more insight into this interaction?

3. The component propagation attack (Attack 3.1) successfully propagated the backdoor for the emotion task but failed completely for the ASR task (AER 0.00%). This is a very interesting and stark contrast. What is the authors' hypothesis for this? Does it suggest that ASR training on clean data is sufficient to "overwrite" the trigger's effect, whereas the emotion task is not?

---

> ### Author Response · Authors · 2025-11-19
>
> We appreciate the thoughtful review. Thank you for the feedback and questions.
>
> **Weaknesses:**
>
> 1. **Limited Scope of Attack:** Our primary goal is to understand how audio backdoors propagate through a cascaded speech language model, such as SpeechLLM. Using a single, relatively simple trigger helps isolate this mechanism without external factors. We expect that even acoustically different triggers would activate the same backdoor mechanism. Regarding clean-label attacks: these methods typically rely on optimization to synthesize highly tailored poisoned examples. While they are powerful, they also introduce an artificial optimization loop that moves us further from our main objective: studying the propagation of an audio backdoor through the pipeline itself. Dirty-label attacks, by contrast, follow a black-box setting and require no additional optimization, making them better suited for isolating this propagation behavior. Clean-label attacks are important and remain promising future work, but they fall outside what we we aim to study here.
>
> 2. **Practicality of Defense:** We agree with this and also highlight the limitations of this defense. Fine-tuning felt like a natural step after seeing the results of Attack 3.1, as it reflects a realistic scenario of reusing a pretrained encoder and adapting it to a new task. Defenses often require additional computation or training, and so may only be applied when there is suspicion of poisoning. We wanted to see if fine-tuning could act as a more “natural” defense, even if the model is poisoned without the user’s knowledge. While we did not explicitly evaluate feature-based defenses (e.g., activation clustering, spectral signatures), the clear separation between clean and poisoned embeddings (Section 7, Figure 2) suggests these approaches could be effective. We highlight this in our revised manuscript (Section 9).
>
> 3. **Single Model Architecture:** Our main contribution is the propagation analysis itself; understanding how and why backdoor signals propagate (or fail to propagate) through each component in a cascaded pipeline. The single-architecture focus was a deliberate choice to enable this: cascaded designs allow clean isolation of each stage (encoder, connector, LLM), making it possible to trace backdoor propagation through each component. Architectures with multimodal fusion would make it difficult to isolate component-level effects. Extending to other architectures is interesting future work, but we believe that mechanistic understanding in a controlled setting is an important first step.
>
> **Questions:**
>
> 1. **Trigger Repetition:** Yes, exactly. We have added an explanation and cited relevant prior work on this phenomenon in Section 6.2 (Trigger Repetition in ASR).
>
> 2. **Fine-tuning Defense:** Yes, we believe our embedding analysis in Section 7 explains this. The backdoor appears subtle and does not compete strongly with clean data, so partial fine-tuning fails to erase it. We attribute this to the inherent uncertainty of the emotion task, which we discuss in detail in Section 7.
>
> 3. **ASR vs Emotion Propagation:** Great intuition, this is exactly what happens. Clean ASR training overwrites the trigger's effect, whereas the emotion task does not. This is also explained in detail in Section 7.

---

> ### Comment · Reviewer_g9Jf · 2025-11-25
>
> Thanks for the detailed response. In general, I think this work has enough contribution on exploring the backdoor attacks on speech language models. I will maintain my positive review to support this work.

---

### Author Response · Authors · 2025-11-19
**Summary of Changes**

As a summary, we have:

- Added a latent space analysis of emotion embeddings vs ASR embeddings for the encoder propagation attack (Attack 3.1) in Section 7. This explains why Attack 3.1 failed for ASR but succeeded for emotion. We are currently running experiments on the remaining tasks (age and gender) to verify if the observed patterns hold. ~~These will be added to the Appendix shortly~~ Results have been added.

- We revised Section 2.3 to highlight the motivation for studying backdoor attacks on speech language models and how they differ from poisoning single speech models or LLMs.

- Explained why ASR poisoning uniquely requires trigger repetition by citing prior work (Section 6.2).

- Defined the explicit threat model (Section 4.1).

- Added an ethics statement.

---

### Note · Authors · 2026-01-03

**Comment:**

Withdrawing to incorporate new results for resubmission.

**Withdrawal Confirmation:**

I have read and agree with the venue's withdrawal policy on behalf of myself and my co-authors.